# Experimental modelling of the growth of tubular ice brinicles from brine flows under sea ice

Sergio Testón-Martínez[1], Laura M. Barge[2], Jan Eichler[1], C. Ignacio Sainz-Díaz[1], Julyan H. E. Cartwright[1,3]

[1]Instituto Andaluz de Ciencias de la Tierra, CSIC--Universidad de Granada, 18100 Armilla, Granada, Spain.
[2]NASA Jet Propulsion Laboratory, California Institute of Technology, 4800 Oak Grove Drive, Pasadena CA, USA
[3]Instituto Carlos I de Física Teórica y Computacional, Universidad de Granada, 18071 Granada, Spain

*Correspondence to*: Julyan Cartwright (julyan.cartwright@csic.es)

**Abstract.**

We present laboratory experiments on the growth of a tubular ice structure surrounding a plume of cold brine that descends under gravity into water with a higher freezing point. Brinicles are geological analogues of these structures found under sea ice in the polar regions on Earth. Brinicles are hypothesized to exist in the oceans of other celestial bodies, and being environments rich in minerals, serve a potentially analogous role as an ecosystem on icy ocean worlds to that of submarine hydrothermal vents on Earth.

## 1 Introduction

Ice brinicles are tubular finger-like formations that grow downwards from the underside of sea ice into the underlying seawater (Cartwright et al. 2013). They are formed when pockets of concentrated salt brine in the sea ice are released (Weissenberger 1992), causing a plume of cold, dense brine to flow downward from the bottom of the ice layer into the surrounding seawater. As the brine descends, the temperature difference between the brine (which, owing to freezing point depression, may reach as low as -23°C while remaining liquid) and the seawater (near its freezing point, ~ -1.8°C) leads to the formation of a hollow tube of ice around the brine that can extend several metres down into the ocean, even sometimes reaching the seafloor (Figure 1). In earlier literature ice brinicles were also termed "ice stalactites" (Paige 1970; Martin 1974; Dayton and Martin 1971; Perovich et al. 1995) but an icicle is an ice stalactite that has a different growth mechanism than a brinicle, so we argue against the use of that term. Brinicles have been found in both the Arctic and Antarctic oceans (Katlein et al. 2020; BBC 2011) and are of interest due to their potential role in the transport of heat, salt, and other materials between the ocean and the atmosphere (Cartwright et al., 2013). Brinicles also share morphological similarities with other self-assembling structures, in particular with chemical gardens (Barge et al. 2015). Both types of structures exhibit a tubular finger-like morphology and can extend several centimetres to several metres in length, and brinicles have been described as an unusual example of an inverted chemical garden (Figure 2) (Cartwright et al. 2013;

Pampalakis 2016). Additionally, both structures are examples of self-organizing systems that exhibit complex dynamical behaviour (Cartwright et al., 2013).

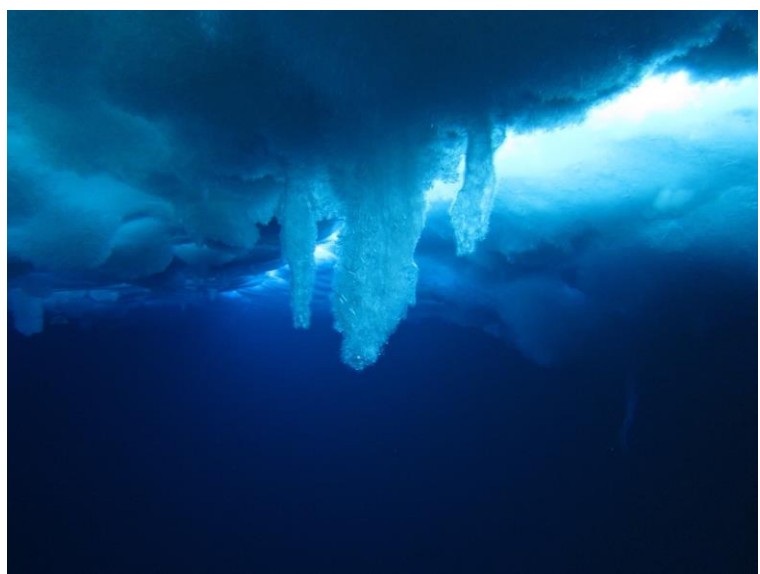

**Figure 1:** Brinicles filmed under sea ice near McMurdo base in Antarctica. Image courtesy of Rob Robbins, US Antarctic Program.


The study of brinicles dates back several decades, with early research focused on their formation, structure, and dynamics. Field observations from Antarctica have been reported from the 1970s on (Paige 1970, Dayton and Martin 1971, Perovich 1995, BBC 2011). A first laboratory study from 1974 used a combination of experiments and theoretical models to investigate the formation and growth of brinicles in seawater (Martin 1974). Martin found that the inner and outer brinicle

radius increases with time, and noted that the interior flow is convectively unstable and controls the brinicle tip radius. Also, as determined by Martin (1974), only a fraction of the ice formed actually contributes to forming the brinicle, and most ice crystals are swept out into the surrounding seawater. As brinicles inject cold brine beneath the ice layer, they may be relevant to ice sheet desalination and upper ocean mixing (Perovich et al. 1995; Dayton and Martin 1971). It is important to consider the potential biological and ecological impacts of ice brinicles, as microbial communities have been discovered in

hypersaline brines (Bougouffa et al. 2013; Steinle et al. 2018). Brinicles may also form in extraterrestrial environments and affect ocean mixing and microenvironments on other celestial bodies, for example under the ice shell of Jupiter's moon Europa (Vance et al. 2019; Buffo et al. 2021).

Although there has been significant progress made in understanding the formation and dynamics of brinicles, there is still much to be learned about these structures: in particular, what conditions lead to their formation, and what effects

variations in these conditions (growth velocity, morphology, and microstructure of the precipitates) have on their formation.

In this study, we created brinicles in the laboratory using different methods in order to study the conditions under which they might form on Earth and on other worlds.

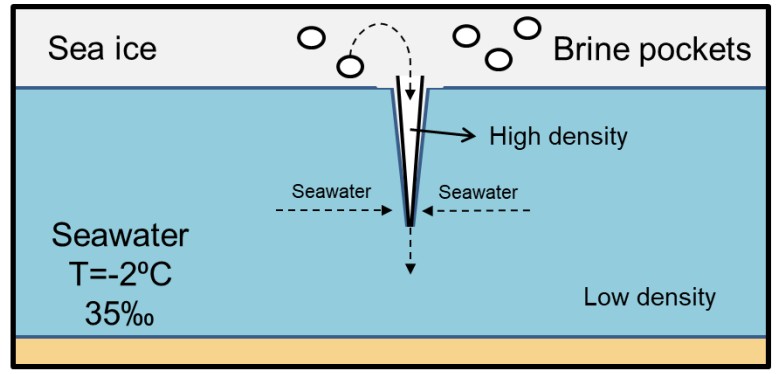

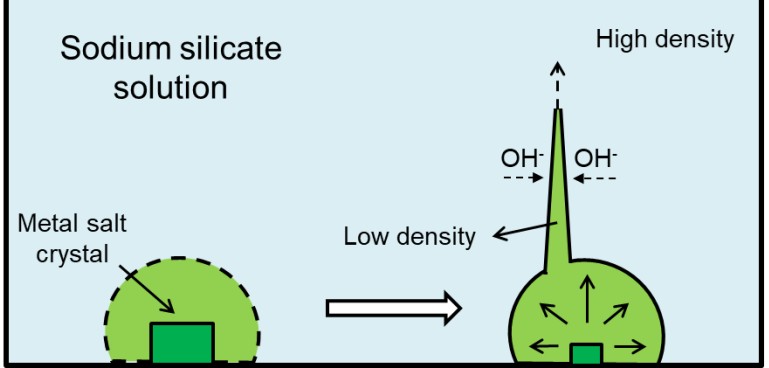

**Figure 2:** (a) Diagram of the formation of a brinicle, illustrating how the brine is ejected downwards from sea ice and freezes the seawater forming a tubular ice structure. (b) Diagram of the formation of a chemical garden, illustrating the creation of the garden membrane, as well as its rupture and tubular growth.

## 2 Methodology

Three different experimental methods were used to attempt to grow brinicles in the laboratory from a cold brine flow into a reservoir of less-salty water. We varied the size of the reservoir in which the brine flow occurred as well as the injection rate and aperture (Table 1).

      The first method – "2D-cell" – used a Hele-Shaw cell (54 x 54 mm), formed by two methacrylate walls with 1 mm of separation between them, as has been used in previous studies to create chemical gardens confined in two dimensions
(Rocha et al. 2021; Ding et al. 2016; Haudin et al. 2014). The second method – "3D cell" – also used a similar Hele-Shaw cell but with a larger (10 mm) separation between the methacrylate walls. We should also note saline water freezing experiments in a Hele-Shaw cell by Niedrauer and Martin (1979) and Middleton et al. (2016). The brine was composed of a 25% weight solution of NaCl in tap water chilled in a -24 °C freezer until completely frozen. Then it was taken out and left defrosting in a -4 °C freezer for two hours before injection. The Hele-Shaw cell or beaker was filled with cold tap water at ~0

℃. In the 2D cell and 3D cell experiments, the brine at ~ -18 ℃ was injected by hand using a 10 mL syringe through a 0.9 mm gauge needle at approximately 0.06 mL/s, 0.1 mL/s, or 0.2 mL/s. (These values were determined dividing the volume injected by the total time of the experiment.) The 2D and 3D cell experiments were conducted in a cold room at 4 ℃, while some of the 3D cell experiments were also carried out at 16 ℃. We also tried to inject the brine using a syringe pump, which offers greater control over the injection rate. However, this was unsuccessful since the ~0.3m of tubing needed to connect the

syringe to the Hele-Shaw cell facilitated too much heat transfer to the brine, even when insulated using packing wrap, such that upon injection the temperature was no longer low enough for a brinicle to form. Experiments were monitored by filming with a Nikon D3400 DSLR camera set up 20 cm from the Hele-Shaw cells. For a 3D cell experiment, we also studied the formation of the structures with a Schlieren optics setup, as in a previous study of salt fingers (Linden 1973) and sea ice applications (Middleton et al. 2016). The same camera and a blue laser were used to witness fluid flow during brinicle

formation.

The third method – "beaker" – used a larger glass rectangular container of 3L. It was initially filled with 2L $H_2O$ + 3.5% mass fraction of NaCl (sea water analogue) cooled down to -1 ℃. The brine was a saturated NaCl solution cooled down to -20 ℃. The brine was injected using a hose with a diameter of 3 mm using a valve to adjust the flow rate, which was important in order to initiate the brinicle growth which started when flow was established. The flow rate was estimated

considering that 1 L of brine took typically 15-20 minutes to flow through the system. The experiments were performed in a cold room at 4 ℃. The brine container and the tubing were thermally isolated using foam material. Experiments were monitored by filming with a Nikon D3400 DSLR camera set up 20 cm from the experiment. Different setups were tried; in later runs there a second tube was added to remove the brine from the bottom of the beaker. In that way we could reduce mixing of sea water and brine and maintain a relatively constant salinity of the sea water. As a result, using this setup, the

brinicle would grow until reaching the bottom of the container.

|  | Temperature of water (℃) | Temperature of brine (℃) | Temperature of the cold room (℃) | Injection rate (mL/s) |
|---|---|---|---|---|
| 2D cell experiment | 0 | -18 | -4 | 0.1 |
| 1st 3D cell experiment | 0 | -18 | 16 | 0.06 |
| 2nd 3D cell experiment | 0 | -18 | 16 | 0.1 |
| 3rd 3D cell experiment | 0 | -18 | 16 | 0.2 |
| Beaker experiment | -1 | -20 | -4 | 1 |

**Table 1:** Experimental data of the temperatures of the water, brine, and cold room used for each experiment in ℃, as well as the brine injection rates for each method.

The 3D cell and beaker experiments formed brinicles, and these were further analysed using photographs taken throughout the experiments. For the 3D cell experiments at the three injection rates, we measured every 5 seconds: the total length of the brinicle; the thickness of the brinicle at the top part that touches the injection needle; and the diameter of the lowest point of the brinicle. With the beaker experiments, videos of brinicle growth were used to make measurements of the same parameters every 5 seconds.

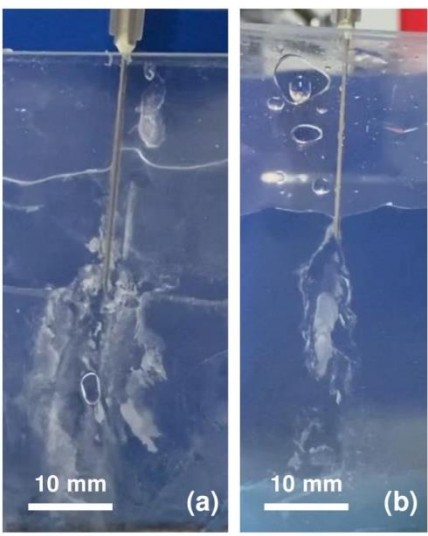

**Figure 3:** Two brinicle experiments (a) and (b) following the 2D cell method with a 1 mm separation between the cell walls, 10 seconds after injection. In both cases solid, instead of tubular, ice structures are formed. See movieS1 in supplementary material.

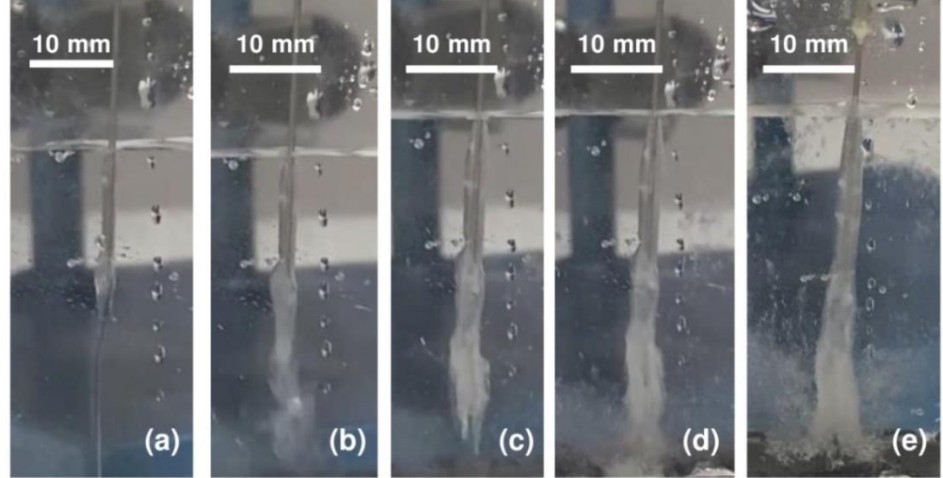

**Figure 4:** Evolution of a brinicle created with the 3D cell method. (a) Start of the injection and freezing of the flow walls (00:03.13). (b) Downward freezing of the brinicle (00:10.10). (c) Widening of the brine flow walls (00:13.13). (d) The brinicle reaches the bottom of the cell (00:16). (e) Freezing of the "seafloor" (00:19.35). See movieS2 in supplementary material.

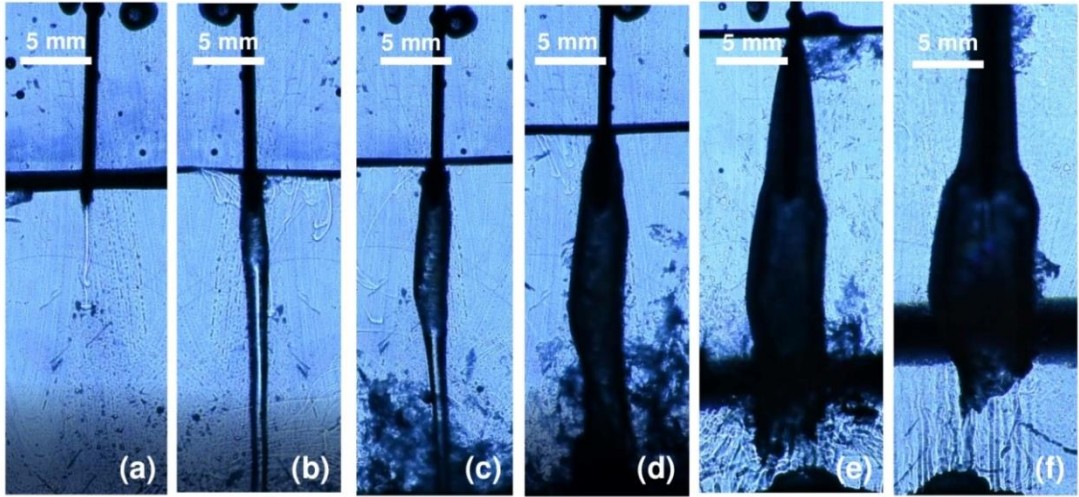

**Figure 5:** Evolution of a brinicle created with the 3D cell method, filmed with a Schlieren optical setup. (a) Start of the injection (00:00). (b) Initial freezing of the brine flow walls (00:07.12). (c) Downwards freezing of the brinicle walls (00:10.80). (d) Freezing at the bottom of the cell and release of ice particles (00:20). (e) Breaking down of the brinicle and ice layer deposition at the interface (00:48.48). (f) After the injection was stopped; melting of the deposited ice and reshaping of the brinicle (01:15.68). See movieS3 in supplementary material.

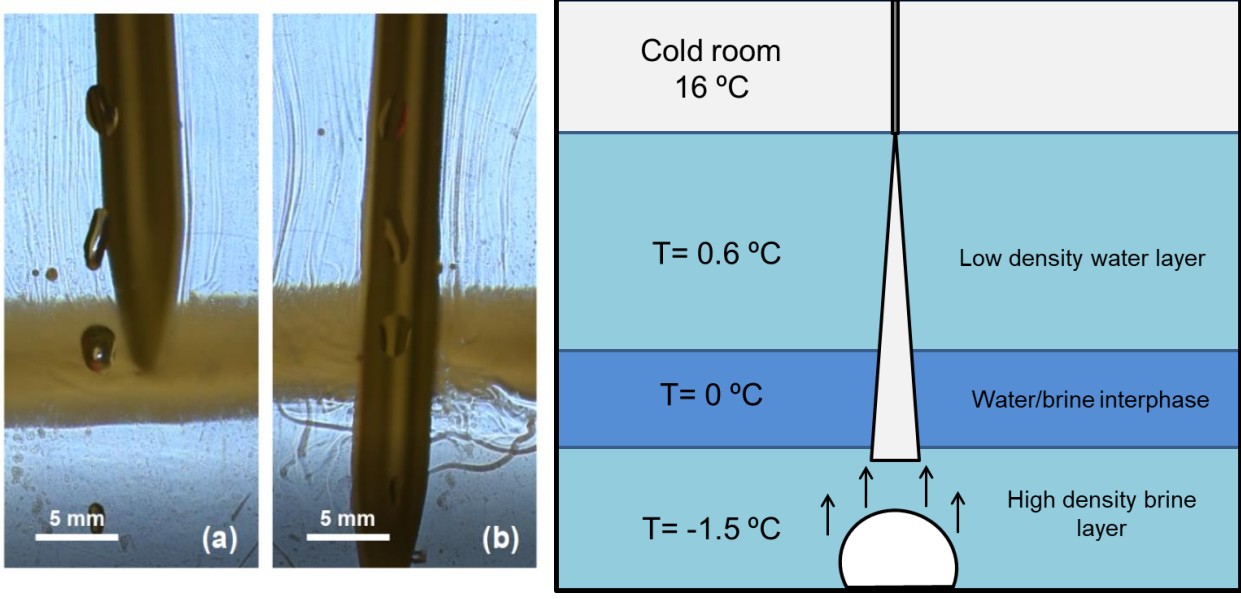

**Figure 6:** Photography and sketch of the interface created during the 3D Hele-Shaw cell experiment in Figure 5. A Schlieren method was used to observe the movement of the fluids when interacting with a temperature probe (a) above the brine/water interface and (b) below. The yellow colour of the interface is a highlighting optic effect created with a lights set.

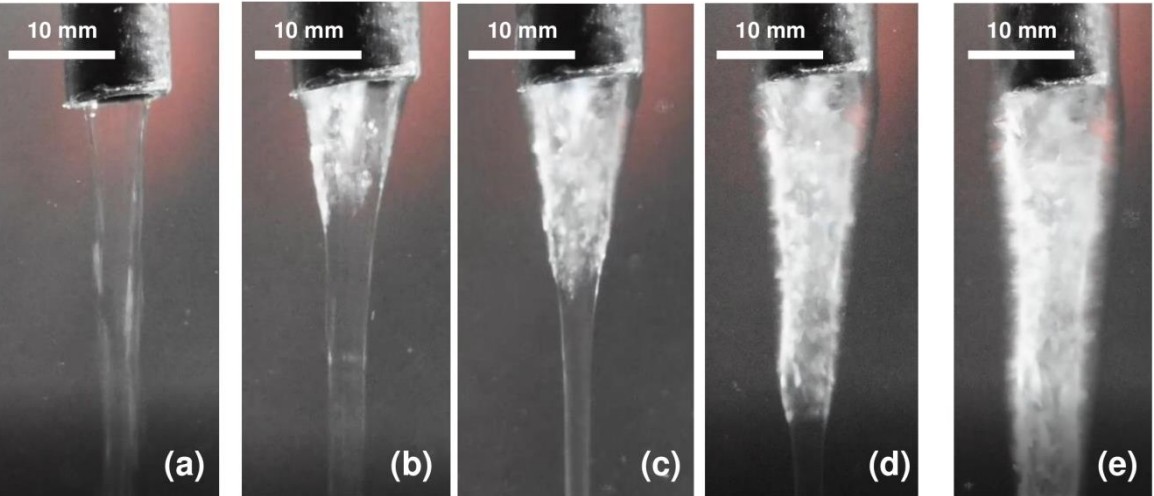

**Figure 7:** Evolution of a brinicle created with the beaker method. (a) Start of injection (00:00). (b) Start of freezing of the brinicle walls (00:07.91). (c) Downwards freezing of the brine flow (00:14.45). (d) Widening of the brinicle over time (00:40.21). See movieS4 in supplementary material.

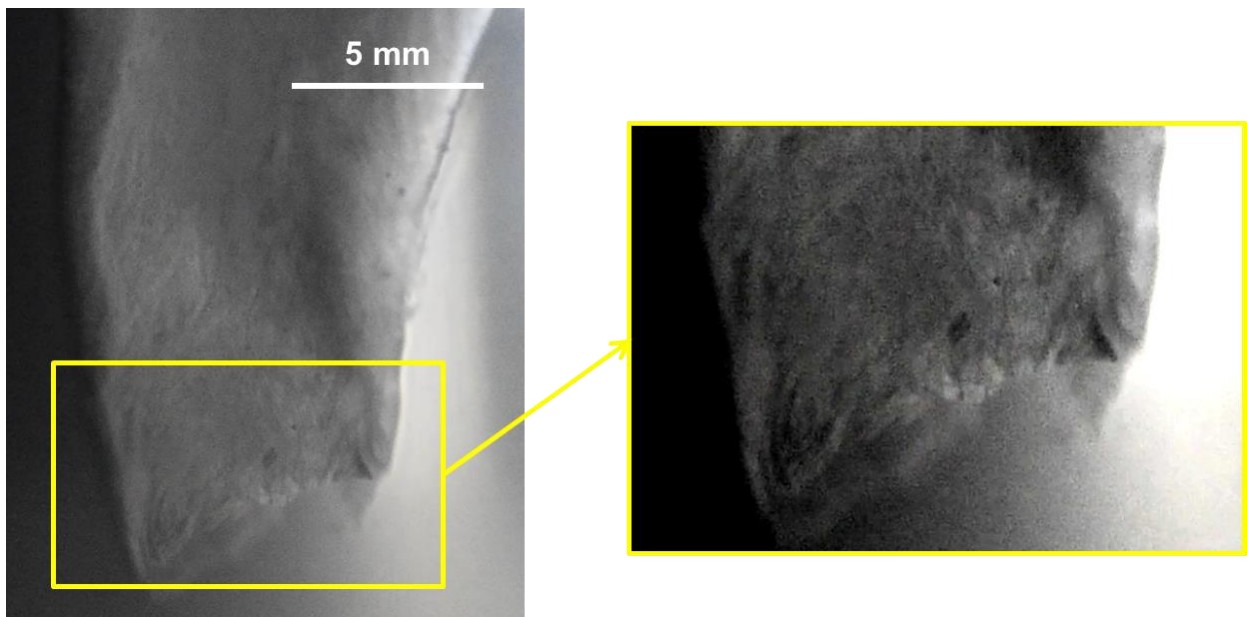

**Figure 8:** Detail of the tip of the Brinicles formed in the beaker, showing its high porosity.

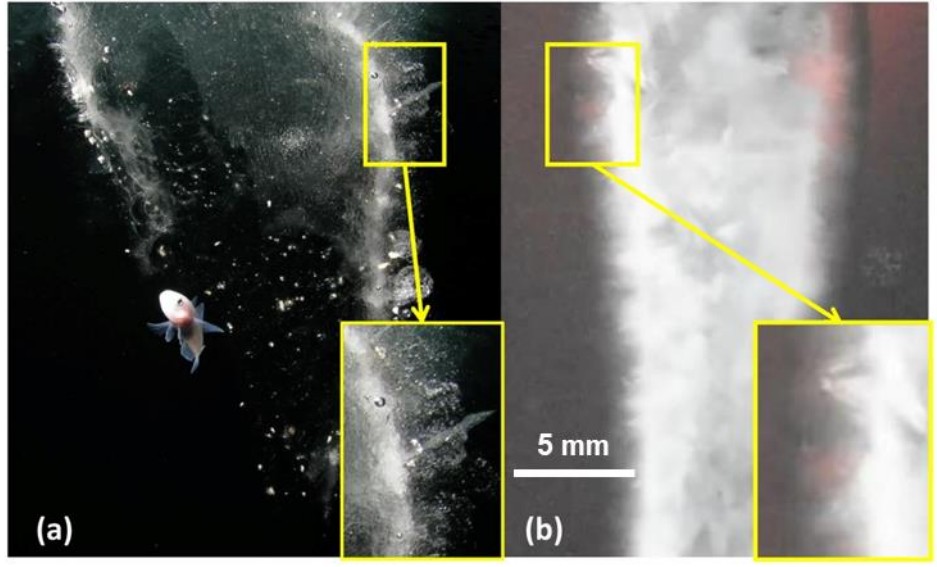

**105**

**Figure 9:** Comparison between the walls of (a) a brinicle found near McMurdo base in Antarctica, and (b) a brinicle made in the laboratory using the beaker method. In both structures horizontally oriented crystals perpendicular to the brinicle wall can be seen. Image (a) is courtesy of Rob Robbins, US Antarctic Program. An approximate scale of (a) is provided by the fish.

## 3 Results

### 3.1 Laboratory brinicle formation

The 2D Hele-Shaw cell method with 1 mm separation, which we have used previously to form injection chemical
gardens of various chemistries (Haudin et al. 2014), did not lead to brinicle formation because the cell width was too narrow for ice tube growth. The narrow separation between the cell walls only facilitated a 2D brine flow structure, rather than a cylinder as in 3D. As shown in Figure 3, upon brine injection the seawater surrounding the brine flow began to freeze and formed a closed ice structure around the needle and brine, impeding the creation of a 3D brinicle. Instead the brine touched the walls and caused ice formation in all directions. After some time, more brine could not be injected because it was
obstructed by ice (see movieS1 in supplementary material).

The 3D cell method formed brinicles, and details of the fluid flows could be observed with a Schlieren setup. With a greater separation between walls (10 mm), the self-clogging phenomenon observed for the 2D cell method did not occur and a tubular ice structure was formed (Figure 4). (See movieS2 in supplementary material). As the brine started to flow into the cell, it sank toward to the bottom due to the density difference, and freezing was observed starting from the top of

the brine flow as the brinicle width and length increased with time. Ice also formed around the brine injection needle because it was colder than the reservoir. Shortly after the start of the injection, the ice wall formed around the brine flow was very thin, as can be observed in Figure 5b; the ice wall thickened over the course of the injection (Figure 5c). Eventually, the brinicle reached the bottom of the cell (Figure 5d) and the brine flow began to accumulate at the bottom forming a bi-layer system with a high density brine phase and the water phase while continuing to freeze at the water/brine interface, forming a

brine layer capped by an ice layer in the lower part of the cell making the deposited ice layer grow thicker (Figure 5e). (See movieS3 in supplementary material). When the brinicle reaches the bottom, the growth of the brine phase causes the hollow brinicle walls to expand and form a bell-shaped tube.

After stopping the injection (Figure 5f), we can observe how the brinicle zone, which is into the brine phase, melts falling ice pieces to the bottom. A series of upward flows are observed due to dilution or freshening of the system (Figure 5f).

Moreover, the fluid interface (black band in Fig 5e) separating the high density phase with the salt concentration from the lighter water phase was tested and the insertion of a temperature probe into this interface (Figure 6) revealed that temperatures varied in different areas of the cell: above the interface it was 0.6 ºC and below the interface (Fig 6b) -1.5/3 ºC, whereas the actual interface (Fig 6a) had a temperature of approximately 0 ºC. In Figure 6 we can also see that the interface (highlighted with a yellowish colour created with a light-set) remained stable when being touched by the temperature probe;

only after stirring the solution thoroughly with the probe, did this band disappear.

The beaker method also formed brinicles, which were larger than those in the 3D cell. Figure 7 shows the formation of a brinicle in the beaker experiment: the brine flow from the hose (Fig. 7a) starts to freeze from the top (Fig. 7b), and continues to freeze the brine flow walls (Fig 7c) while widening its diameter (Fig. 7d) until the bottom of the container (Figure 7e). (See movieS4 in supplementary material). The completed brinicle had textured ice walls that appeared porous, as seen in the

image in Figure 8. In the larger brinicles formed in the beaker experiment, we observed that on the outer edges of the brinicle there were horizontally oriented ice crystals visible (Figure 9); this was also observed in photos of brinicles from Antarctica (Figure 9) where the initial ice crystals also grew perpendicular to the brine flow. These crystals are probably oriented normal to the *c* axis, along the *a* axis, known to be the fastest growth direction in ice (Glen and Perutz 1954).

In all experiments, brinicles once formed were very fragile; it was not possible to physically extract them from the

experiments and if the needle or hose was moved they would quickly disaggregate.

### 3.2 Physical analysis of brinicle growth

In the 3D cell experiments, the brinicles' growth velocities, calculated by measuring the change in the brinicle height every 5 seconds, decreased over time, and moreover, brinicles grew more slowly at lower brine injection rates (Figure 10a). These data fit the model from Martin (1974), who analysed brinicle growth in terms of a Graetz problem — the fluid

mechanics of tube flow that couples fluid flow with heat transfer — that brinicle growth is proportional to the square root of time. As the brinicle's length increases with time, when the brinicle is long enough, the brine that arrives at the lowest point

is not as cold as the initial brine injected because it receives heat from the surroundings. This makes the freezing of the brinicle and consequent growth more difficult than at the start.

The widening of the brinicles at the starting point was observed to slow down with time (Figure 10b), probably due to the brinicle walls being too thick at some point to facilitate further freezing on the outside of the wall. We chose to measure the top of the brinicle that touches the needle because it is frozen from the start so the widening is more obvious. All the experiments at different injection rates have a similar tendency in width increase, indicating that this widening is independent of the brine flux.

Figure 10c represents the diameter of the lowest point of the brinicle at that time point. As shown in Figure 4, the brinicle walls grow thicker with time; this is reflected in the observation that the lowest diameter of a brinicle also becomes wider as the brinicle grows. To confirm this, we also measured the brinicle width at one of the lowest points for a time lapse (at the same brinicle height) after its creation, and we found that it follows the same tendency shown in Figure 10b where its width increase grows and then slows down. The flux of brine also affects this behaviour, as seen in Figure 10c where the experiments with a higher flux have a faster growth than the ones with lower flux. The oscillations visible in the curves imply an oscillatory growth instability that warrants further investigation. This may be related to the periodic popping regime observed in chemical gardens (Barge et al 2015; Thouvenal and Steinbock, 2003).

The brinicle formed in the beaker experiment (Figure 7) was similarly measured using one of the videos (see movieS4 of the supplementary material) and showed similar behaviour to the brinicles in the 3D cell experiments (Figure 4): the brinicle growth velocity decreases with time (Fig. 10a); while the width of the brinicle at the injection point increases until it becomes stable (Fig. 10b); and the thickness of the lowermost part of the brinicle increases with time (Fig. 10c).

Brinicle and chemical gardens are both instances of fluid-jet-templated tube growth; they differ in that brinicles grow by thermal diffusion, which is approximately 100× faster than the chemical diffusion in chemical gardens (Cardoso and Cartwright 2017). Nevertheless, the same physical model can be brought to bear on them. This shows that the tube radius should vary with flow rate following Poiseuille flow driven by a pressure gradient $-\mathrm{d}P/\mathrm{d}z$ and a density difference $\Delta\rho_i g$ (Cardoso and Cartwright 2017),

$$Q_i = \left(\Delta\rho_i g - \frac{dP}{dz}\right)\frac{\pi R^4}{8\mu_i} \tag{1}$$

Where R is the radius and $\mu_i$ the viscosity. This is shown in Figure 11.

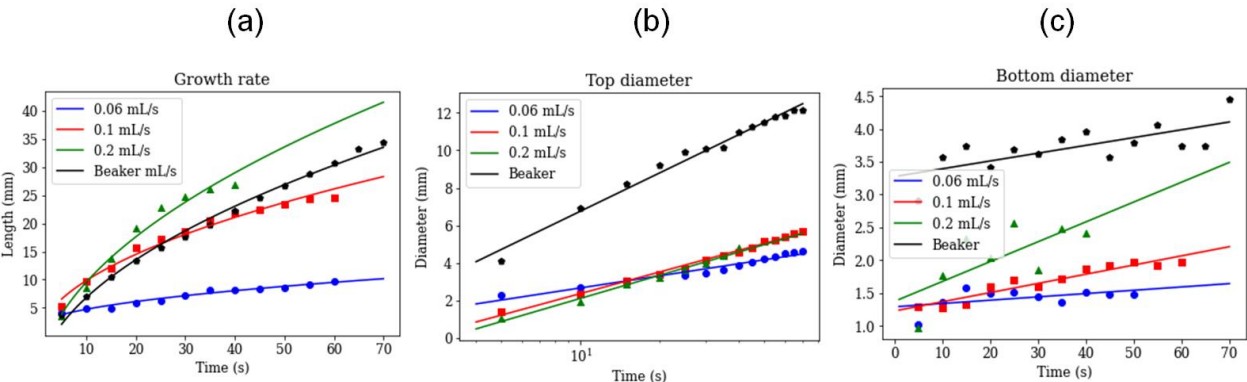

**Figure 10:** Measurements of (a) brinicle growth rate, (b) brinicle diameter at the brine injection point, and (c) at the bottom over time. The first three lines represent a 3D cell experiment with different brine flow rates: 0.06 mL/s (blue), 0.1mL/s (red), and 0.2 mL/s (green); while the last one (black) used the beaker method. The fit in figure (a) is proportional to the square root of time (Martin 1974). An error of ~1% (10 μm) comes from the distance measuring program. Note that the last points in each series represent the brinicle touching the bottom of the container, causing them not to follow the expected fit. See movieS5 and movieS6 in supplementary material.

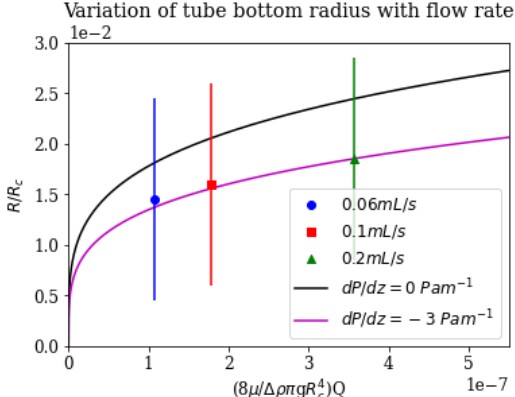

**Figure 11:** Brinicle measurements of the radius at the bottom of the brinicle at the same time for different flow rates. The three points represent a 3D cell experiment with different brine flow rates: 0.06 mL/s (blue), 0.1mL/s (red), and 0.2 mL/s (green). The first fit line (black) represents the prediction from Poiseuille flow (Cardoso and Cartwright 2017) if dP/dz=0 Pa m$^{-1}$, and the second if dP/dz=-3 Pa m$^{-1}$. An error of ~1% (10 μm) comes from the distance measuring program. See movieS5 and movieS6 in the supplementary material.

## 4 Discussion

Growth of brinicles in a laboratory setting is highly dependent on the parameters of the experimental setup, each of

which reflect particular aspects of a natural brinicle system. In this work the most high-fidelity analogues to natural brinicles

were formed in a large glass container with brine fed through a hose; in the smaller, more controlled Hele-Shaw cell systems,

a larger separation between the walls than in previous work was required to witness brinicle growth. This is consistent with other experimental studies that have used similar setups to simulate growth of ice. In previous studies, Hele-Shaw cells with 2-3 mm separation have been used to simulate ice growth and formation of brine channels, in which planar freezing fronts were observed but no brinicles, even when brine channels formed (Middleton et al. 2015; Eide and Martin 1975). In our 1 mm "2D cell" experiment we also observed freezing fronts that touched the cell walls but this separation was too narrow to permit ice tube formation. The size and shape of the brine flow aperture is also important, as it determines the initial brinicle diameter (Fig. 10) as well as the brine flow rate. In a natural system, where brine flow into seawater is the result of many connected brine pockets and channels within the ice (Weissenberger 1992), the radius of the brine channel draining into the seawater below would not be constant, as in our experiments where needles and hoses are used, but would adjust to the brine flow rate as more buoyant seawater intrudes up into the neck and freezes, until the force of seawater buoyancy and brine pressure are balanced (Eide and Martin 1975). In our experiments, the injection (0.06-0.2 mL/s) which takes the place of the force of brine pressure is still lower than estimated brine flow rates in natural brinicles, ~ 1 mL/s to 16 mL/s, reported by Perovich et al. (1995) and Dayton and Martin (1971) respectively, dictated by the amount of brine flowing downward from underneath the sea ice.

Temperature is a crucial parameter, since for brinicles to form, seawater must be near freezing point (< = 1.8°C), while the brine temperature must be much lower (< -15°C). A rise in temperature in either or both fluids will make brinicle formation impossible as the seawater does not get cold enough to freeze. This was one of the lessons learned in our cell experiments: use of a syringe pump would be preferred in terms of controlling the experimental parameters; however, the requirement of tubing to connect the syringe to the cell led to too much temperature change in the brine before it could contact the water reservoir. To optimize the flow rate and injection aperture position, while maintaining the sub-freezing brine temperature, the whole experiment should be done with a syringe pump in a cold room.

Brinicles grow relatively quickly; in our experiments brinicles centimeters to 10's of cm long formed in minutes, which is comparable to growth speeds witnessed in geological environments; natural brinicle growth speeds have been reported from ~0.2 – 2 cm / minute (Perovich et al. 1995; Dayton and Martin 1971). The size, spacing, and internal structures of brinicles can vary as well, depending on their growth conditions. Both small (10 – 100cm long) (Paige 1970) and large (1–6m long, 5–25cm wide) (Dayton and Martin 1971; Perovich et al. 1995, Paige 1970) brinicles have been observed in McMurdo Sound, Antarctica. The brinicles in our 2D cell experiments appear to have one single channel (Figure 5), due to brine injection from a single narrow injection point, although we were not able to remove them from the esperimental apparatus to analyse their internal structure. However, in natural systems, brinicles can be more heterogeneous in structure; e.g. Perovich et al. (1995) reported a brinicle from Antarctica that had five separate internal sub-channels. Brinicles occur alone or in groups, but they occur sparsely (~ 5-10 meters spacing between them; Paige 1970, Perovich et al. 1995). This is because there is a minimum amount of seawater necessary to generate the concentrated brine (estimated 10 L seawater per L of brine; Cartwright et al. 2013), and, depending on the size of the brinicle and the brine channel from which

it formed, Dayton and Martin 1971 and Perovich et al. 1995 estimated 130 L and 70 L of brine respectively was needed to grow brinicles that were observed in the field.

    The extent to which brinicles could be relevant for the origin of life and for astrobiology (Vance et al. 2019; Cartwright et al. 2013; Buffo et al. 2021b, Howell & Leonard 2023; Lawrence et al 2023; Howell et al 2024) would depend on how long they could persist in Earth or other planetary ocean-ice interfaces. Modeling studies have suggested that

brinicles could form under the ice shell on Jupiter's moon Europa, and that their lifetime would be dependent on the ice shell thickness – giving a range of hours to years (Buffo et al. 2021) which encompasses the lifetimes of brinicles in terrestrial systems (e.g. in Antarctica brinicles can persist for months, Dayton and Martin 1971). Saturn's moon Enceladus also has a sub-surface ocean and putative hydrothermal activity, and its plumes offer the opportunity to sample the subsurface ocean (Hsu et al. 2015; Postberg et al. 2009); it is possible that brinicles could form in Enceladus as well, though as yet there is no

published modelling on brinicles in an Enceladus context.

    Brinicles are initially fragile, and in our experiments, we observed that moving the injector after the experiment caused the brinicle to disaggregate. Similar fragility has been described for brinicles in the field which disintegrated when touched by a remotely operated vehicle arm (Perovich et al. 1995), though in that study they were able to recover the top 750 cm of a brinicle for analysis. Other brinicles were described as "fragile but flexible enough to bend slightly with gentle tidal

currents" (Paige 1970). The lifetime of a brinicle would therefore depend on the currents which would affect the stability of structure of the brine plume under the ice.  Brinicles photographed under the Antarctic ice are often long and thin; they are high aspect ratio tubes.  They are fragile, easily broken off, and – the argument goes (Buffo et at 2021) – would break off with larger currents or would be prevented from forming at all due to dispersion of the brine plume, and so would not be of importance in an icy world ocean. But brinicles are not always so long and thin; for example, the central one in Figure 1 is

short and wide. If a brine stream descends from sea ice into a lateral ambient current, it will have a different form to a plume in a quiescent fluid medium (Slawson and Csanady 1967, Hewett et al 1971).  Hewett, Fay and Hoult (1971) discuss the basic scaling length for a buoyant plume in a cross flow, which they term the buoyancy length $l_B$,

$$l_B = \frac{gQ_h}{\pi \rho_i C_p T v^3}$$

(2)

where $Q_h$ is the heat flux in the plume, $g$ the gravitational acceleration, $\rho_i C_p$ the heat capacity per unit volume of effluent

fluid (i.e., the outflowing brine), $T$ the temperature of the ambient fluid (i.e., the ocean) and $v$ the cross-flow speed. $l_B$ is approximately the radius of curvature near the stack exit of a buoyant plume of negligible initial momentum. This gives a first-order estimate of the length a brinicle might grow to in a cross-flow situation.

Figure 12 shows this behaviour for brinicles on Earth as well as on Europa and Enceladus, two icy ocean moons. The buoyancy length of a plume, and so of a brinicle decreases fast, with the cube of the speed of a transverse current. A brinicle

in this environment on an icy world might look something like some seafloor hydrothermal vents that form in places with

ambient currents. The ice making up such a brinicle would probably become more massive and less porous as the structure ages, somewhat similar to how snow falling on Earth gradually forms firn and then ice under aging processes (Bartels-Rausch et al 2012). Brinicles on Earth occur in a range of environments including shallow waters (e.g., BBC, 2011) but also have been observed in water up to 300 m deep (Dayton and Martin 1971); they also have been observed to occur far from land in the Arctic (Katlein et al. 2020). Therefore, we propose that brinicles may exist in some form even with currents in icy-world oceans. Since brinicles represent sites of high thermal and chemical gradients at the ice-ocean interface, they could be of interest as locations that might provide energy to support prebiotic chemistry or life (Vance et al 2019; Cartwright et al. 2013). Further studies of these fascinating ice brinicle systems may shed light on the conditions that are required for their formation in different ocean–ice chemistries, and their possible stability and lifetime on Earth and other celestial bodies.

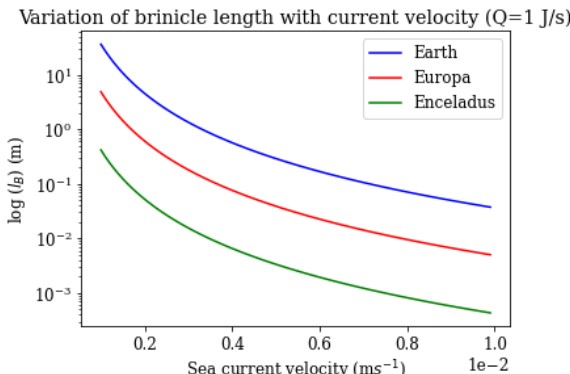

**Figure 12:** Brinicle length calculated using the scaling length for a buoyant plume in a cross flow (equation 2) for different sea current velocities. The length has been calculated using general parameters for Earth (blue), Europa (red), and Enceladus (green) for a fixed value $Q_h$ of the heat flux in the plume.

**Data availability:** All data are available in the paper and the supplementary videos at https://doi.org/10.20350/digitalCSIC/15364.

— *movieS1_2D_cell.mp4*: Movie of the growth of brinicles in a 2-D Hell_Shaw cell. Brine injection rate of 0.1 mL/s. The water freezes in all directions.

— *movieS2_3D_cell.mp4*: Movie of the growth of brinicles in a 3-D Hell_Shaw cell. Brine injection rate of 0.1 mL/s. A tubular ice structure (brincle) is formed.

— *movieS3_3D_Schlieren.mp4*: Movie of the growth of brinicles in a 3-D Hell_Shaw cell using the Schlieren optics technique. Brine injection rate of 0.1 mL/s. Water, brine and ice flows can be seen. Brinicle grows until the freezing the bottom, then the flow stops.

— *movieS4_Baker.mp4*: Movie of the growth of brinicles in a 3-D beaker reactor. Brine injection rate of 1 mL/s. The brine frezes the water creating a brinicle that grows downwards and sideways.

— *movieS5_3D_cell_0,1mls.mp4*: Movie of the growth of brinicles in a 3-D Hell_Shaw cell. Brine injection rate of 0.1 mL/s. The water around the needle freezes and a brinicle starts to grow downwards. Lower speed and diameter than with higher flows.

— *movieS6_3D_cell_0,2mls.mp4*: Movie of the growth of brinicles in a 3-D Hell_Shaw cell. Brine injection rate of 0.2 mL/s. The water around the needle freezes and a brinicle starts to grow downwards. Higher speed and diameter than with lower flows.


**Author contributions:** All authors contributed in an integrated fashion to this work.

**Competing interests:** The authors declare that they have no conflict of interest.

**Acknowledgements**: We thank Rob Robbins for providing the Antarctica brinicle photographs, which have not been previously published. L.M.B. was supported by JPL Topical Research & Technology Development and a NASA PECASE

award; L.M.B's research was carried out at the Jet Propulsion Laboratory, California Institute of Technology, under a contract with the National Aeronautics and Space Administration (80NM0018D0004). The authors would like to acknowledge the contribution of the European COST Action CA17120 supported by the EU Framework Programme Horizon 2020. S.T. acknowledges the CSIC and Spanish Andalusian 'Garantía Juvenil' project AND21_IACT_M2_058.

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
