# Peer review of "Experimental modelling of the growth of tubular ice brinicles from brine flows under sea ice"

_The Cryosphere, 2023_

## Referee Comment (RC1)

**Review of manuscript tc-2023-100**

by Sönke Maus

This is a review of the manuscript *Experimental modelling of the growth of tubular ice brinicles from brine flows under sea ice* by S. Teston-Martinez et al.

Below I cite from the Cryosphere Discussions manuscript tc-2023-100 in *italic font*.

**I   Summary**

The paper presents results from laboratory experiments on growth of tubular ice brinicles that form when high-salinity brine is injected into solutions close to the salinity and freezing point of sea water. Such brinicles have been observed under natural sea ice and have been studied in a couple of previous studies. As the role that such brinicle may play in the earth system or regional ecosystems is not well known, laboratory studies of these features are of interest in sea ice research. The experiments by the authors indicate that brinicles do not form in very thin Hele Shaw Cells and therefore have not been observed in such experiments. For the 3D setups where brinicle growth was achieved the authors present some interesting results and imagery. However, the quantitative results could be presented better in terms of existing models, and the few equations given by the authors are not properly described or tested versus observations. While the manuscript is interesting and generally well written, I suggest to improve it along the following lines.

I. Laboratory settings. From the short descriptions it is not clear under which conditions the different experiments have been performed. I recommend to add a table that gives an overview. Helpful would be a table that provides the essential information for the three experimental setups (e.g. temperature of water and injected brine, injection rates, temperature of cold room, etc...).

II. The work by Martin (1974) is mentioned in several sections. The paper could be improved by outlining some of Martins model equations in relation to the present work. The two equation (1) and (2) given by the authors are insufficiently explained and it is not clear how they are related to the presented results.

III. Many figures are missing scale bars and experimental time (for the time series) and could be improved.

IV. Considerable parts from the 'Discussion' section should better be presented in a 'Results' section, and these too sections should thus be restructure. Focusing on the actual results rather than discussing hypothesis that cannot be tested by the data would also improve the paper.

**II  Specific comments**

**1. Abstract**

L11–14 *We present laboratory experiments on the growth of a tubular ice structure surrounding a plume of cold brine that descends under gravity into water with a higher freezing point. Brinicles are geological analogues of these structures found under sea ice in the polar regions on Earth.* –> Not clear to me - it sounds as if brinicles are natural analogues of tubular ice structures produced in the lab. I would say that both are brinicles.

L14 *They may be important for the energy budget of sea ice.* –> Or they may not. The paper does not present an analysis of this question. I suggest to remove this sentence from the abstract and rather focus on some paper results.

**1. Introduction**

L17 *Ice brinicles* –> When and by whom was this term used for the first time?

L21–58  *-20 ℃* –> -23 ℃  is sufficiently well known for seawater.

L60–L91 –> The experimental description is not clear (e.g. chilling to -24 ℃, then defrosting at -4 ℃, then brine was injected at -18 ℃). Helpful would be a table that provides the essential information for the three experimental setups (e.g. temperature of water and injected brine, injection rates, temperature of cold room, etc...

L70 –> *brinicles have been described as an unusual example of an inverted chemical garden (Figure 2)* –> Fig. 2 does not show or say anything about chemical gardens.

Fig. 2 –> Could be inproved by giving typical brine and ice temperature + salinity during observations of brinicle formation.

**1. Methodology**

L57–58 *as has been used in previous studies to create chemical gardens confined in two dimensions (Rocha et al. 2021; Ding et al. 2016; Haudin et al. 2014).* –> The chemical garden is an interesting analogy, but not the first to mention here (and it was already mentioned in the introduction). More relevant references in comparison are the saline water freezing experiments in a Hele Shaw cell by Niedrauer and Martin (1979) and Middleton et al. (2016).

L60–L91 –> The experimental description is not clear (e.g. chilling to -24 ℃, then defrosting at -4 ℃, then brine was injected at -18 ℃). Helpful would be a table that provides the essential information for the three experimental setups (e.g. temperature of water and injected brine, injection rates, temperature of cold room, etc...

L70 –> *Schlieren optics setup, as in a previous study of salt fingers (Linden 1973)* –> There are many applications of Schlieren optics in fluid dynamics, but here rather the

sea ice applications, like Middleton et al. (2016) should be referenced to.

Fig. 4, 5 and 7 –> The times should be indicated for these series.

Fig. 8 –> *Detail of the tip of the Brinicles formed in the beaker, showing its high porosity* –> What shows the high porosity in this figure?

Fig. 8 and 9 –> Scale bars are missing.

**3.Discussion**

L115-153 –> This part should rather be presented in a Results than in a Discussion section.

L129–132 *Eventually, the brinicle reached the bottom of the cell (Figure 5d) and the brine flow began to accumulate at the bottom forming a bi-layer system with a high density brine phase and the water phase while continuing to freeze at the water/brine interface, forming a brine layer capped by an ice layer in the lower part of the cell making the deposited ice layer grow thicker (Figure 5e).*–> The bottom evolution is not that clearly seen in the figure - a drawing could help to illustrate it better.

L135 *brane* –> brine

L136–137 *A series of upward flows are observed from the melting of this ice because of the rise in temperature* –> I suspect this flow is rather due to dilution/freshening.

L137 *and this time the ice and the contact with the brine solution* –> Seems incomplete - something lacking here?.

L150–151 *These crystals 150 are probably oriented along the a axis, known to be the fastest growth direction in ice.* –> a- or other axis have not been defined or mentioned so far. Give a reference. I also suggest 'normal to the c-axis'.

L157 *These data fit the model from Martin (1974) that brinicle growth is proportional to the square root of time.* –> The basic model features from Martin could be given here and discussed quantitatively in relation to the experiments. Which are the properties and mechanisms that govern the growth velocity?.

L167-168 *As shown in Figure 9b, the brinicle walls grow thicker with time;* –> How can this growth be observed in a single figure?

L172-174 –> *The oscillations visible in the curves imply an oscillatory growth instability that warrants further investigation. This may be related to the periodic popping regime observed in chemical gardens (Barge et al 2015; Thouvenal and Steinbock, 2003).*–> As the oscillations are observed at the bottom of a growing brinicle they more likely present a spatial oscillation. The pattern that form in such a freezing system can be understood in terms of brine release, constitutional supercooling and morphological stability (Mullins and Sekerka, 1964; Hardy and Coriell, 1973; Sekerka, 1967).

Eq. (1) and Figure 11 –> The terms in the figure are not explained in the text or equation, e.g.: What is $R_c$? How is $dP/dz$ determined in the experiment, and how is $\Delta\rho_i$ computed? Also helpful would be to discuss how Martin (1974) has incorporated Poiseuille flow in his model framework.

Eq. (2) –> This is the second equation given, yet it is not tested at all in this work and remains rather hypothetic, and its relationship to the present observations or other previous work remains unclear. E.g., what heat flux $Q_h$ does the equation refer to? At least some plausible values from the field or laboratory should be used with Eq. (2) to estimate the length of a brinicle.

L277-283 –>...*Therefore, we propose that brinicles may exist in some form even with currents in icy-world oceans.*–> There are indeed a couple of observations, but there is no support in observations that brinicles would survive or form with currents. The recent paper by Katlein et al. (2000) only notes 'occasional' observations of brinicles during the year-long MOSAiC expedition. Hence, without giving evidence in the present paper this hypothesis should be left out or rather formulated as a question and motivation for future work, for example: At which levels of under-ice flow and turbulence may brinicles form? How to design laboratory experiments to answer this question?

**References**

Hardy, S.C., Coriell, S.R., 1973. Surface tension and interfacial kinetics of ice crystals freezing and melting in sodium chloride solutions. J. Cryst. Growth 20, 292–300.

Martin, S., 1974. Ice stalactites: Comparison of laminar flow theory with experiments. J. Fluid Mech. 63, 51–79.

Middleton, C.A., Thomas, C., Wit, A.D., Tison, J.L., 2016. Visualizing brine channel development and convective processes during artificial sea-ice growth using schlieren optical methods. J. Glaciol. 62, 1–17. doi:10.1017/jog.2015.1.

Mullins, W.W., Sekerka, R.F., 1964. Stability of a planar interface during solidification of a dilute binary alloy. J. Appl. Phys. 35, 444–451.

Niedrauer, T.M., Martin, S., 1979. An experimental study of brine drainage and convection in young sea ice. J. Geophys. Res. 84, 1176–1186.

Sekerka, R.F., 1967. A time-dependent theory of satbility of a planar surface during dilute binary alloy solidification, in: Peiser, H.S. (Ed.), Crystal Growth, Pergamon Press, Boston, 20-24 June 1966. pp. 691–702.

---

## Author Response (AR1)

Dear Sergio,

First of all, I would like to apologize a long delay of the review process. That was just due to problems to obtain two experts reviews. The both reviewers suggest only a minor revision. Please, provide a revised version according to your plans.

As an editors, I have a one comment. Your additional videos are really interesting but could you add a brief description on experiment for each video.

Best wishes,
Jari Haapala

We added the following descriptions both in the repository and in the supplementary information data at the end of the manuscript:

*movieS1_2D_cell.mp4*: Movie of the growth of brinicles in a 2-D Hell_Shaw cell. Brine injection rate of 0.1 mL/s. The water freezes in all directions.

*movieS2_3D_cell.mp4*: Movie of the growth of brinicles in a 3-D Hell_Shaw cell. Brine injection rate of 0.1 mL/s. A tubular ice structure (brincle) is formed.

*movieS3_3D_Schlieren.mp4*: Movie of the growth of brinicles in a 3-D Hell_Shaw cell using the Schlieren optics technique. Brine injection rate of 0.1 mL/s. Water, brine and ice flows can be seen. Brinicle grows until the freezing the bottom, then the flow stops.

*movieS4_Baker.mp4*: Movie of the growth of brinicles in a 3-D beaker reactor. Brine injection rate of 1 mL/s. The brine frezes the water creating a brinicle that grows downwards and sideways.

*movieS5_3D_cell_0,1mls.mp4*: Movie of the growth of brinicles in a 3-D Hell_Shaw cell. Brine injection rate of 0.1 mL/s. The water around the needle freezes and a brinicle starts to grow downwards. Lower speed and diameter than with higher flows.

*movieS6_3D_cell_0,2mls.mp4*: Movie of the growth of brinicles in a 3-D Hell_Shaw cell. Brine injection rate of 0.2 mL/s. The water around the needle freezes and a brinicle starts to grow downwards. Higher speed and diameter than with lower flows.